# Associations between Optimism and Attentional Biases as Measured by Threat-Avoidance and Positive-Search Tasks

**DOI:** 10.3390/healthcare11040617

**Published:** 2023-02-19

**Authors:** Michio Maruta, Suguru Shimokihara, Yoshihiko Akasaki, Yuma Hidaka, Yuriko Ikeda, Gwanghee Han, Goro Tanaka, Toshio Higashi, Takefumi Moriuchi, Takayuki Tabira

**Affiliations:** 1Department of Occupational Therapy, School of Biomedical Sciences, Health Sciences, Nagasaki University Graduate, Nagasaki 852-8520, Japan; 2Faculty of Medicine, Kagoshima University, 8-35-1, Sakuragaoka, Kagoshima 890-8544, Japan; 3Graduate School of Health Sciences, Kagoshima University, 8-35-1, Sakuragaoka, Kagoshima 890-8544, Japan; 4Tarumizu Municipal Medical Center, Department of Rehabilitation, Tarumizu Chuo Hospital, 1-140 Kinko-cho, Tarumizu 891-2124, Japan; 5Department of Rehabilitation, Okatsu Hospital, Medical Corporation, Sanshukai, 3-95, Masagohonmachi, Kagoshima 890-0067, Japan; 6Department of Occupational Therapy, Faculty of Medicine, School of Health Sciences, Kagoshima University, 8-35-1, Sakuragaoka, Kagoshima 890-8544, Japan; 7Department of Occupational Therapy, School of Health Sciences at Fukuoka, International University of Health and Welfare, Okawa 830-8501, Japan

**Keywords:** optimism, attentional bias, dot-probe task, threat avoidance, emotional visual search task, positive search

## Abstract

Evidence suggests that optimism has a positive impact on health status. Attentional bias modification (ABM) may be beneficial for enhancing optimism, but its effective application requires a detailed investigation of the association between attentional bias and optimism. This study aimed to determine the association between attentional bias and optimism based on different task types. Eighty-four participants completed the attentional bias measures using the dot-probe task (DPT), emotional visual search task (EVST) paradigms, and psychological assessments. Optimism was assessed using the Life Orientation Test-Revised with subscales for optimism and pessimism. Pearson’s correlation coefficient and multivariate linear regression analysis were applied to investigate the association between optimism and attentional bias. Neither the attentional bias derived from DPT nor EVST was significantly correlated with optimism total score or subscales. Regression analysis also showed no association between attentional bias and optimism (DPT, β = 0.12; EVST, β = 0.09), optimism subscales (DPT, β = 0.09; EVST, β = 0.17), or pessimism subscales (DPT, β = −0.10; EVST, β = 0.02). Our findings showed no evidence that attentional biases derived from either the DPT or EVST measures are associated with optimism or pessimism. Further studies are needed to effectively adapt the ABM to enhance optimism.

## 1. Introduction

There is growing evidence that optimism plays an important role in the successful aging of older adults [1]. Optimism is a psychological trait defined by the general expectation that good rather than bad things will happen in one’s future [2]. Being optimistic promotes positive health behaviors in older adults and may provide independent health benefits to them [3,4]. Indeed, many studies have found that optimistic older adults have better functional ability [5,6] and subjective well-being [7] and lower mortality [3] and cardiovascular disease risk [8] than those who are not optimistic. Further, the health and longevity benefits of optimism may not be dependent on race or ethnicity [9]. The importance of optimism as a protective factor for mental health is similar in social contexts such as the ongoing COVID-19 pandemic [10,11,12]. Optimism is a modifiable factor and may provide a point of intervention to improve health before the occurrence of an adverse health-related event in older adults [6,13].

In recent years, attention bias modification (ABM) has emerged as a potentially effective intervention for enhancing optimism [14]. The ABM is a computer-based cognitive training paradigm designed to retrain dysfunctional biases in thinking. Numerous studies have reported that ABM improves symptoms associated with anxiety [15], depression [16], and chronic pain [17] by shifting emotional biases toward more positive and less negative stimuli. Most research on ABM has used the dot-probe paradigm, which avoids attention from negative stimuli, whereas positive-search ABM paradigms developed a focus on searching for positive stimuli. As many studies have shown that using positive-search ABM biases people’s attention away from negative to positive social information, it may be appropriate as a useful approach to enhance people’s optimistic state. Indeed, Kress et al. suggested that performing positive-search ABM training with repeated attention to positive stimuli while ignoring negative stimuli enhances optimism [14]. Additionally, ABM training has the potential to be widely used because it requires few resources and can be adapted for online applications. However, the state of attentional bias before and after training has not been assessed, and the association between optimism and attentional bias remains unclear. Furthermore, given the underlying mechanisms of attentional bias, which vary by measurement method [18], its association may differ with optimism toward positive or negative future events (e.g., optimism for positive future events is associated with attentional bias derived from positive search, whereas optimism for negative events is associated with attentional bias derived from threat avoidance). The association between optimism and attentional bias needs to be investigated to develop effective ABM training for enhanced optimism.

This study aimed to examine whether attentional bias, as measured by the dot-probe and positive-search paradigms, is associated with optimism. We hypothesized that optimism and attentional bias are associated and that their association depends on the type of task that derives attentional bias (i.e., threat-avoidance tasks are associated with pessimism, while positive-search tasks are associated with optimism). Clarifying these relationships may contribute to the development of effective attentional bias training for enhancing optimism.

## 2. Materials and Methods

### 2.1. Participants

A priori power analysis using G*power software was performed to calculate an adequate sample size for correlation analysis. An effect size of 0.3, a significance level of 0.05, and a power of 0.8 were considered. Eighty-four samples were estimated. We recruited healthy Japanese volunteers from the rehabilitation staff at the hospitals to which the authors (Y.A. and Y.H.) belong and from undergraduate and graduate students at Kagoshima university and Nagasaki University. A total of 87 healthy subjects were enrolled in this study. All the participants had normal or corrected-to-normal vision. None of the participants had a history of neurological or psychiatric disorders. The study was approved by the Ethics Committee on Epidemiological Studies of Kagoshima University (Ref No. 210273) and conducted in accordance with the Declaration of Helsinki. Written informed consent to participate in this investigation was obtained from all participants.

### 2.2. Measures

#### 2.2.1. Optimism

Optimism was assessed using the Life Orientation Test-Revised (LOT-R), which has been validated in Japanese samples [19]. The LOT-R consists of 10 items, with optimism and pessimism assessed with 3 items, respectively, and the remaining 4 items are filler. Participants rated their level of agreement with each question on a 5-point Likert scale ranging from 0 (“strongly agree”) to 4 (“strongly disagree”). The total score for optimism was calculated by reversing the score for negative phrasing (pessimism), with total scores ranging from 0 to 24. We used subscales of optimism (0–12) and pessimism (0–12) as well as total optimism scores [20]. Higher scores reflect more optimistic or pessimistic dispositions.

#### 2.2.2. Attentional Bias Measurement

##### Apparatus and Stimuli

Attentional bias was measured using the dot-probe task (DPT) and emotional visual search task (EVST) [21,22,23]. The apparatus and visual stimuli were identical in the two attentional bias measurement tasks. The attentional bias measurement tasks, i.e., DPT and EVST, were programmed using E-prime 3.0 (Psychology Software Tools, USA) and presented on a 14-inch laptop computer (CF-LV9, Panasonic, Osaka, Japan) with a screen refresh rate of 60 Hz (1920 × 1080 resolution). Participants were seated approximately 45 cm away from the screen, and their responses were collected via mouse clicks. The facial expressions used as stimuli in this study were obtained from the facial expression database of the National Institute of Advanced Industrial Science and Technology (AIST), Japan [24]. Permission to use the facial expression database was obtained through the AIST.

##### Dot-Probe Task

The DPT adapted the facial stimulus-based task from the Tel Aviv University National Institute of Mental Health Attention Bias Measurement Toolbox, which was designed to measure standardized attentional bias (https://people.socsci.tau.ac.il/mu/anxietytrauma/research/, accessed on 21 November 2022). We replaced the facial stimuli with images obtained from the Japanese database to adapt to the sample in this study. The DPT comprised 120 trials: 80 threat–neutral and 40 neutral–neutral trials. In the threat–neutral trials, pairs of threat and neutral facial expressions were randomly presented at the top and bottom of the screen, respectively. In the neutral–neutral trials, pairs of neutral facial expressions were randomly presented at the top and bottom of the screen. The trials began with a black fixation cross presented for 500 ms, after which a threat–neutral or neutral–neutral pair was presented for 500 ms. Following the removal of facial images, a probe (“<” or “>”) appeared in either the same location as the threat face (“congruent trials”) or that of the neutral face (“incongruent trials”). Participants were required to click the left or right mouse button in the direction of the probe. The probe was presented until the response was recorded, and the participants were instructed to respond as quickly and accurately as possible. Ten practice trials were conducted before the assignment to ensure that participants performed correctly.

Reaction times (RT) for threat–neutral trials were used to calculate the attentional bias index. The attentional bias index was calculated by subtracting the mean RT of congruent trials from that of incongruent trials. Consistent with standard practice, trials with incorrect responses and extremely short (<150 ms) and extremely long (>2000 ms) RTs were excluded [21,23,25]. Trials of RTs that were outside ±2 standard deviation (SD) of the participant’s mean for threat–neutral trials were also excluded. Higher scores indicate a stronger negative bias.

##### Emotional Visual Search Task

The EVST comprised two separate blocks of positive and negative trials, each with 32 trials [21,22]. Participants were required to repeatedly select either the happy face from a 4 × 4 grid of negative (anger, fear, sadness) faces (positive trials) or only the negative face from a 4 × 4 grid of happy faces (negative trials). The trials began with a black fixation cross presented for 500 ms, after which a 4 × 4 grid of facial expression images appeared. The grid of faces was presented until the response was recorded, and the participants were instructed to respond as quickly and accurately as possible. The participants responded to each trial by clicking directly on the target location using a computer mouse. The cursor was reset to the center of the computer screen at the beginning of each trial. The order of the positive or negative trials was counterbalanced across participants. Three practice trials of each type were conducted before the assignment to ensure that the participants performed correctly.

The attentional bias index was calculated by subtracting the mean RT of the negative trials from that of the positive trials. Consistent with standard practice, trials with incorrect responses and extremely short RTs (<200 ms) were excluded [22,23]. Trials of RTs outside ±2 SD of the participant’s mean for each of the two conditions were also excluded. Higher positive scores reflected more interference from negative information.

#### 2.2.3. Other Variables

We also assessed participant characteristics, including age (years), sex, education (years), mood, rumination, and personality traits, as potential factors influencing optimism. Participants’ affect was assessed using the Positive and Negative Affect Schedule (PANAS) [26]. The PANAS is a self-reported questionnaire comprising 20 items, with 10 items each for positive and negative affect. The total score ranges from 10 to 60 points for positive and negative affect, respectively. Higher scores reflect higher positive and negative affect, respectively. Rumination was assessed using the rumination subscale of the Rumination–Reflection Questionnaire (RRQ) [27]. The RRQ is a self-reported questionnaire comprising 24 items, 12 of which assess rumination. All items were rated on a 5-point Likert scale ranging from 1 (strongly disagree) to 5 (strongly agree), with higher scores reflecting higher levels of rumination. The personality traits of participants were assessed using the Japanese version of the Ten Item Personality Inventory (TIPI-J) [28,29]. The TIPI-J measures five domains originating from the Big Five theory (i.e., extraversion, agreeableness, conscientiousness, neuroticism, and openness). The TIPI comprises 10 items, including two items for each of the five personality traits. Each item is rated on a 7-point Likert scale ranging from 1 (strongly disagree) to 7 (strongly agree), and each personality trait is rated on a scale of 2–14 points. Higher scores reflect a higher trait.

### 2.3. Procedure

After signing the informed consent form, participants performed two attentional bias measurement tasks: the DPT and EVST. The order of the DPT or EVST tasks was counterbalanced across participants. After the completion of the attentional bias measurement task, the participants were required to complete four self-reported questionnaires. The assessment took approximately 45 min.

### 2.4. Statistical Analysis

Participant characteristics are presented as the mean and standard deviation for each assessment. A one-sample t-test was used to calculate whether the mean attentional bias of all the participants was significantly different from zero. Pearson’s correlation coefficient was used to investigate the association between optimism, attentional bias, and other variables. Attentional biases were also examined in correlation with optimism and pessimism subscales. In addition, we explored the association between optimism and attentional bias using multivariate linear regression analysis adjusted for demographic and psychological variables.

All statistical analyses were performed using R version 4.1.2, with a significance level set at 5%.

## 3. Results

### 3.1. Characteristics of the Study Participants

Ultimately, 84 participants (36 males and 48 females; mean age, 24.6 ± 4.5 years; mean education, 15.6 ± 1.7 years) were included in the analysis, except those with missing psychological assessment data (*n* = 3). Table 1 summarizes the participants’ characteristics. For the DPT, incorrect trials and trials with RTs more than two SDs from the individual’s mean were removed (6.3%). A one-sample t-test showed no evidence of attentional bias derived from the DPT (*p* = 0.585). For EVST, incorrect trials and trials with RTs more than two SDs from the individual’s mean were removed (0.9%). A one-sample t-test showed that the participants exhibited an attentional bias toward positive stimuli (*p* = 0.007).

### 3.2. Association between Optimism and Attentional Biases

Figure 1 illustrates the correlation between the total optimism score and the optimism and pessimism subscales and attentional bias indices. The attentional bias derived from DPT was not significantly correlated with any of the optimism measures (total score: *r* = 0.052, *p* = 0.641; optimism subscale: *r* = 0.144, *p* = 0.192; pessimism subscale: *r* = 0.068, *p* = 0.536). The same was observed for attentional bias derived from the EVST (total score, *r* = 0.045, *p* = 0.685; optimism subscale, *r* = −0.002, *p* = 0.988; pessimism subscale, *r* = −0.063, *p* = 0.536).

### 3.3. Association between Optimism and Each Psychological Assessment

Table 2 summarizes the correlations between optimism and each psychological assessment. Positive affect (*r* = 0.451, *p* < 0.001), extraversion (*r* = 0.314, *p* = 0.004), and openness (*r* = 0.444, *p* < 0.001) showed significant positive correlations with optimism, whereas rumination (*r* = 0.438, *p* < 0.001) and neuroticism (*r* = 0.271, *p* = 0.013) showed significant negative correlations with optimism.

Table 3 summarizes the results of the multivariate linear regression analysis. Attentional biases derived from either the DPT (total score: *β* = 0.12, *p* = 0.22; optimism subscale: *β* = 0.09, *p* = 0.39; pessimism subscale: *β* = −0.10, *p* = 0.39) or EVST (total score: *β* = 0.09, *p* = 0.36; optimism subscale: *β* = 0.17, *p* = 0.09; pessimism subscale: *β* = 0.02, *p* = 0.86) task showed no association with optimism total scores, optimism subscale, or pessimism subscale. There were significant associations between positive mood (*β* = 0.30, *p* = 0.02), rumination (*β* = -0.36, *p* = 0.01), and openness (*β* = 0.26, *p* = 0.04) on the optimism total score. In addition, rumination (*β* = −0.28, *p* = 0.01) and openness (*β* = 0.29, *p* = 0.03) showed significant associations with the optimism subscale, and rumination (*β* = 0.35, *p* < 0.01) showed a significant association with pessimism subscale.

## 4. Discussion

This study investigated whether attentional biases derived from two different measures are associated with optimism. Our findings showed no evidence of a cross-sectional association between attentional bias and optimism in either task. 

We hypothesized that attentional biases would be differentially associated with optimism or pessimism depending on the characteristics of the measurement task; however, contrary to expectations, we could not confirm their association. The attentional biases derived from the DPT and EVST are associated with the degree of anxiety and depression symptoms [23]. Meanwhile, in a review of ABMs for anxiety, the possibility of deriving emotional benefits independent of pre- and post-training attentional bias status was noted [30]. Additionally, cognitive bias modification with the repetitive interpretation of positive imagery increases optimism; however, even in that study, it was not accompanied by changes in cognitive bias [31]. Therefore, the impact of cognitive bias modification involving attention may not depend on the presence or degree of cognitive (e.g., attention) bias toward negative information, and our results may support this. However, we cannot address this because this study only assessed the status of attentional bias and did not examine the changes over time. We need to investigate changes in attentional bias before and after implementing ABM to examine the association between attentional bias and optimism in detail.

The possibility that this result may differ depending on the measure of optimism should be considered. This study used the LOT-R, a widely used measure of trait optimism, to assess both optimism and pessimism. Previous studies have shown that ABM does not increase the state of optimism, showing effects only on comparative optimism related to social comparisons [14,32]. They stated that ABM may, specifically, affect comparative optimism because it enhances self-esteem [14], which plays an important role in social comparison [33]. Future studies should include and explore different optimism dimensions.

We also examined affect, rumination, and personality traits as potential factors influencing optimism. We found that positive affect, rumination, and personality traits including extraversion, neuroticism, and openness were significantly correlated with optimism. Multivariate linear regression analyses suggest that positive affect, rumination, and openness are particularly important factors in optimism. Consistent with the results of previous studies [34,35,36,37], our findings confirm the need to consider these variables in future studies investigating the association between attentional bias and optimism.

This study has several limitations. First, this study excluded individuals with psychiatric disorders, but the psychological assessment of depression was not available, so potential depressive effects cannot be ruled out. However, the inclusion of important factors of depression such as rumination and neuroticism as covariates and the very weak association between optimism and each attentional bias in the correlation and linear regression analyses means that the impact on the primary results is unlikely to be significant. Second, this study applied a reaction-time-based measure of attentional bias, which has been the focus of many studies to date. However, attentional bias for negative stimuli derived from reaction times, such as the DPT, has been noted to have poor test–retest reliability [38,39] and internal reliability [38,39,40]. Future studies should apply more direct methods of assessing attention (e.g., eye tracking [40,41]) to examine the association with optimism. Third, although this study was conducted with healthy adults as a basic investigation into the development of methods to enhance optimism, the “positivity effect” should be kept in mind for application to older adults. There is a well-known age-related “positivity effect” in which older adults show an increased preference for positive information over negative information in attention and memory [42]. Considering this, the association between optimism and attentional bias may show different results depending on age. Future studies that expand the age range of the subjects are warranted. Despite these limitations, our findings may have important implications for the development of ABM to increase optimism.

## 5. Conclusions

Our findings showed no evidence that attentional biases derived from either the DPT or EVST measures are associated with optimism or pessimism. Further studies are needed to effectively adapt the ABM to enhance optimism.

## Figures and Tables

**Figure 1 healthcare-11-00617-f001:**
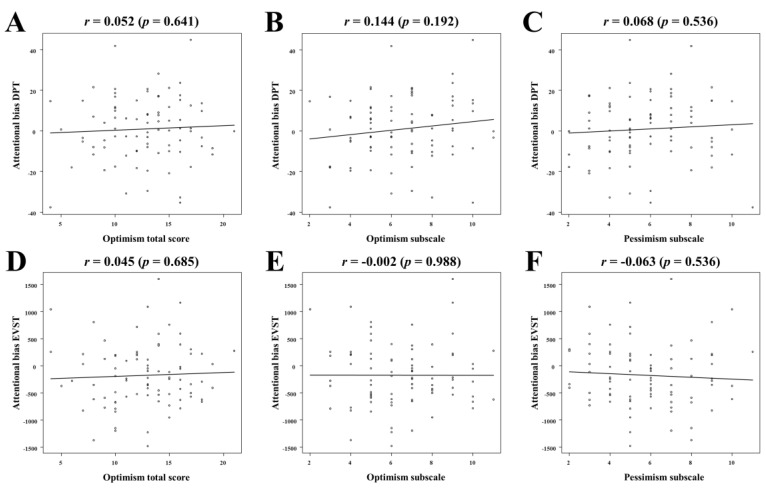
Correlation between optimism and attentional biases. Correlation between attentional bias derived from the DPT and optimism total score (**A**), optimism subscale score (**B**), and pessimism subscale score (**C**). Correlation between attentional bias derived from the EVST and optimism total score (**D**), optimism subscale score (**E**), and pessimism subscale score (**F**). DPT, dot-probe task; EVST, emotional visual search task.

**Table 1 healthcare-11-00617-t001:** Means, standard deviations, and minimum and maximum scores of the attentional bias measurement tasks and the psychological assessments.

		Mean	SD	Min.	Max.
LOT-R	Optimism total score	12.70	3.63	4.00	21.00
	Optimism subscale score	6.54	2.11	2.00	11.00
	Pessimism subscale score	5.80	2.17	2.00	11.00
DPT	RT neutral	439.59	55.16	351.87	616.56
	RT threat	438.65	52.49	359.00	613.42
	Attentional bias index	0.94	15.77	−37.76	44.85
EVST	RT positive	2454.25	716.65	1312.57	4839.30
	RT negative	2627.93	504.84	1440.97	4444.48
	Attentional bias index	−173.68	579.56	−1483.41	1600.44
PANAS	Positive affect	31.11	7.78	15.00	51.00
	Negative affect	22.29	7.19	11.00	47.00
RRQ	Rumination subscale	39.83	7.96	16.00	58.00
TIPI-J	Extraversion	9.21	3.00	3.00	14.00
	Agreeableness	9.85	2.08	5.00	14.00
	Conscientiousness	6.38	2.58	2.00	13.00
	Neuroticism	8.73	2.74	3.00	14.00
	Openness	8.20	2.48	2.00	14.00

SD, standard deviation; LOT-R, the Life Orientation Test-Revised; DPT, dot-probe task; RT, reaction time; EVST, emotional visual search task; PANAS, the Positive and Negative Affect Schedule; RRQ, the Rumination–Reflection Questionnaire; TIPI-J, the Japanese version of the Ten Item Personality Inventory.

**Table 2 healthcare-11-00617-t002:** Correlation between optimism and psychological assessments.

	Optimism	Positive Affect	Negative Affect	Rumination	Extraversion	Agreeableness	Conscientiousness	Neuroticism	Openness
Optimism	1	0.451 **	−0.071	−0.438 **	0.314 **	0.120	−0.037	−0.271 *	0.444 **
Positive affect		1	0.148	−0.215 *	0.441 **	0.148	0.207	−0.187	0.476 **
Negative affect			1	0.269 *	−0.116	−0.015	−0.018	0.327 **	−0.086
Rumination				1	−0.137	−0.180	−0.214	0.465 **	−0.276*
Extraversion					1	−0.101	0.115	−0.104	0.426 **
Agreeableness						1	0.208	−0.221 *	0.207
Conscientiousness							1	−0.203	0.101
Neuroticism								1	−0.460 **
Openness									1

Pearson’s correlation analysis, * *p* < 0.05, ** *p* < 0.01.

**Table 3 healthcare-11-00617-t003:** Association between optimism and attentional biases and psychological variables.

	Optimism Total Score ^a^					Optimism Subscale ^b^					Pessimism Subscale ^c^				
	β	SE	95% CI		*p*	β	SE	95% CI		*p*	β	SE	95% CI		*p*
			Lower	Upper				Lower	Upper				Lower	Upper	
Attentional bias with DPT	0.12	0.00	0.00	0.00	0.22	0.09	0.00	0.00	0.00	0.39	−0.10	0.00	0.00	0.00	0.39
Attentional bias with EVST	0.09	0.02	−0.08	0.06	0.36	0.17	0.01	0.00	0.05	0.09	0.02	0.01	−0.03	0.03	0.86
Age	−0.05	0.09	−0.21	0.14	0.68	−0.15	0.06	−0.18	0.04	0.20	−0.12	0.06	−0.18	0.06	0.36
Sex (ref: female)	**0.27**	**0.85**	**0.27**	**3.65**	**0.02**	0.20	0.52	−0.21	1.88	0.12	−0.24	0.57	−2.19	0.09	0.07
Education	0.02	0.23	−0.42	0.49	0.88	0.06	0.14	−0.21	0.35	0.60	0.05	0.15	−0.25	0.36	0.70
Positive affect	**0.30**	**0.05**	**0.03**	**0.24**	**0.01**	0.17	0.03	−0.02	0.11	0.17	−0.27	0.04	−0.15	0.00	0.04
Negative affect	0.12	0.05	−0.05	0.17	0.27	0.02	0.03	−0.06	0.07	0.86	−0.17	0.04	−0.12	0.02	0.17
Rumination	**−0.36**	**0.05**	**−0.26**	**−0.07**	**0.00**	**−0.28**	**0.03**	**−0.13**	**−0.02**	**0.01**	**0.35**	**0.03**	**0.03**	**0.16**	**0.00**
Extraversion	0.07	0.13	−0.18	0.35	0.52	0.11	0.08	−0.09	0.24	0.35	−0.05	0.09	−0.21	0.15	0.71
Agreeableness	0.00	0.17	−0.33	0.32	0.98	−0.01	0.10	−0.21	0.19	0.92	−0.03	0.11	−0.25	0.19	0.80
Conscientiousness	−0.18	0.14	−0.53	0.03	0.08	−0.12	0.09	−0.27	0.07	0.26	0.21	0.09	−0.01	0.37	0.06
Neuroticism	−0.08	0.16	−0.41	0.21	0.52	−0.10	0.10	−0.27	0.12	0.45	0.02	0.11	−0.19	0.23	0.88
Openness	**0.26**	**0.17**	**0.03**	**0.72**	**0.04**	**0.29**	**0.11**	**0.03**	**0.46**	**0.03**	−0.16	0.12	−0.37	0.10	0.25

SE, standard error; CI, confidence interval; DPT, dot-probe task; EVST, emotional visual search task. General linear model regression analysis with optimism total score, optimism subscale, and pessimism subscale as dependent variables, respectively. ^a^ adjusted R^2^ = 0.38, overall model test; F = 4.86, *p* < 0.001; ^b^ adjusted R^2^ = 0.30, overall model test; F = 3.67, *p* < 0.001; ^c^ adjusted R^2^ = 0.21, overall model test; F = 2.71, *p* < 0.001; Note: significant findings *p* < 0.05 are highlighted in bold.

## Data Availability

The data that support the findings of this study are available from the corresponding author upon reasonable request.

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
