# Peer review of "Associations between Optimism and Attentional Biases as Measured by Threat-Avoidance and Positive-Search Tasks"

_healthcare, 2023, doi:10.3390/healthcare11040617_

Round 1

Reviewer 1 Report

The authors aimed to determine any relationship between optimism and attentional bias as assessed by DPT and VEST paradigms, and psychological scores. The study is properly designed, however the following issues need to be clarified before publication:

-Did participants suffer from depression? this is an important aspect, that needed assessment at least by psychological tools such as the Beck's depression inventory, an that would potentially affect the results.

-Please provide demographic characteristics of participants (age, sex, education), this needs to be clarified and incorporated as factors in the analysis.

-Why a correlation analysis is informative about the linear relationship between variables, it does not reveal information about other more complex relationships.  Please perform a GLM regression analysis by also including the effect of demographic variables in addition to the considered ones.

-

Reviewer 2 Report

Dear authors, thank you very much for your research. I am going to make a number of comments below to improve your research and help you. Add hypotheses and describe them in the results and discussion, if they have been fulfilled or not. In participants put number of men and women and average age, and if you have it, their socio-cultural or educational level. The rest of the methodology is fine. I liked the results very much. In the discussion I see very well explained the limitations but I miss a section on practical applications.
